# Update on Emeralds from Kagem Mine, Kafubu Area, Zambia

**Ran Gao** [1,2], **Quanli Chen** [1,2,3,*], **Yan Li** [1,2] and **Huizhen Huang** [1,2]

1. Gemological Institute, China University of Geosciences, Wuhan 430074, China; gaoran@cug.edu.cn (R.G.); yanli@cug.edu.cn (Y.L.); huizhenhuang_0920@163.com (H.H.)
2. Hubei Province Gem & Jewelry Engineering Technology Research Center, Wuhan 430074, China
3. School of Jewelry, West Yunnan University of Applied Sciences, Baoshan 679100, China
* Correspondence: chenquanli_0302@163.com; Tel.: +86-183-2722-5368

**Abstract:** Kagem emerald mine in Zambia is deemed to the largest open-pit emerald mine with extremely high economic value and market share in the world. To meet the market demand for tracing the origin of emeralds, 30 emeralds from the region were tested, and some discoveries were made compared to previous studies. This study provides a full set of data through standard gemological properties, inclusions, color characteristics, advanced spectroscopic and chemical analyses, including Raman, micro micro-UV-Vis-NIR, FTIR, and LA-ICP-MS. The most common inclusions in Kagem emeralds are two-phase inclusions, which exhibit elongated, hexagonal, oval, irregular shapes or appear as negative crystals with incomplete hexagonal prism. These inclusions consist of $H_2O$ or $H_2O + CO_2$ (liquid) and $CO_2 + N_2$ or $CO_2 + N_2 + CH_4$ (gas). Mineral inclusions typically include actinolite, graphite, magnetite, and dolomite. Black graphite encased in actinolite in Kagem emeralds is first reported. The FTIR spectrum of Kagem emeralds reveals that the absorption of type II $H_2O$ is stronger than that of type I $H_2O$, indicating the presence of abundant alkali metals, which was confirmed through chemical analysis. Kagem emeralds contain high levels of Na (avg. 16,440 ppm), moderate-to-high Cs (avg. 567 ppm), as well as low-to-moderate levels of K (avg. 185 ppm) and Rb (avg. 14 ppm) concentrations.

**Keywords:** emerald; Kagem Mine; gemological characteristics; origin traceability



## 1. Introduction

Emerald is the green variety of beryl colored by chromium and vanadium, and its ideal chemical formula is $Be_3Al_2SiO_{18}$. The traditional classification system for emerald deposits has been expanded upon by Giuliani et al. (2019) [1]. Emerald occurrences and deposits were reclassified into two main types: tectonic magmatic-related (Type I) and tectonic metamorphic-related (Type II), and further subdivided into seven sub-types based on the host rock types. The Kafubu deposit in Zambia was classified as Type IA, associated with mafic-ultramafic rocks.

Zambia is one of the most significant emerald sources worldwide, second only to Colombia. Kagem emerald mine is located in the Ndola Rural Emerald Restricted Area and lies south of Kitwe and west of Ndola, in Zambia's Copperbelt Province (Figure 1). Kagem Mine is considered the world's largest single open-pit emerald mine, accounting for approximately 25%–30% of global emerald production, with a potential mine life of 22 to 2044 years [2]. Besides the Kagem Mine, there are other larger mines in the Kafubu area, including Kamakanga, Grizzly, and Chantete [3,4].

Bank (1974) [5] proposed that the necessary chromic element in emeralds, chromium, comes from the magnetite in talc-magnetite schist. Koivula (1982) [6] was the first to report on the presence of tourmaline as inclusion in Zambian emeralds. Sliwa & Nguluwe (1984) [7] described the geological setting of Zambian emeralds. Seifert et al. (2004a) [8] first provided quantitative geochemical, petrological, and mineralogical data on the major rock types of the Kafubu emerald deposits. Seifert et al. (2004b) [9] conducted a comprehensive

study of the environmental impact of the emerald mines in the Kafubu area. Zwaan et al. (2005) [3] first reported the Musakashi area, a new emerald deposit initially worked by local miners in 2002 [10]. In 2014, GIA Lab conducted a field exploration of the history, region geology, mining methodology, processing, operation, and auction of the Kagem Mine and published a detailed report [11]. However, detailed studies on the inclusion, spectroscopy, and chemical composition of Kagem emerald still need to be completed [12].

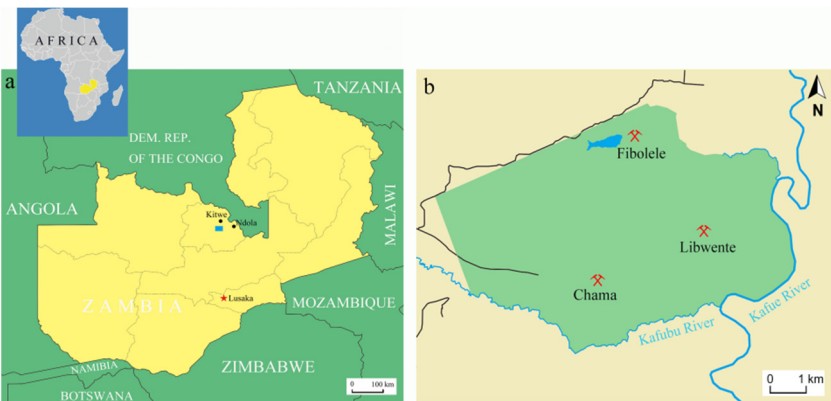

**Figure 1.** (**a**) Zambia is a landlocked country located in the southern part of Africa, the capital is Lusaka. Kagem Mine (blue rectangle) is located in north-central Zambia, near the Kafubu River in the Ndola Rural Emerald Restricted Area (NRERA); (**b**) There are three operating pits in the Kagem Mine: Chama, Libwente, and Fibolele. Picture: Ran Gao.

We received a batch of emerald samples from the Kagem Mine, Zambia in September 2022. This article provides a brief overview of the history and region geology of the Kagem Mine and a detailed research of gemological properties, inclusions, UV-Vis-NIR, FTIR, and chemical composition analysis. These analyses can effectively differentiate our materials from those sourced from other significant emerald localities.

## 2. Geological Setting

The Kagem emerald concession covers an area of approximately 46 square kilometers within a 200-square-kilometer productive zone and comprises the current operating Chama open pit mine and the bulk sampling pits at Libwente and Fibolelem [3].

The Kafubu area is located geologically at the center of the transcontinental Pan-African belts in central-southern Africa. The Crustal evolution is thought to be related to three successive orogenic events: the Ubendian, Irumide and Lufilian (Pan-African) [13–15]. The Kafubu emerald deposits occur within metamorphic rocks of the Muva Supergroup that date back to around 1700 Ma [million years ago] [16,17]. Muva Supergroup overlays the basement granite gneisses, consisting of footwall mica schist, talc-magnetite schist, amphibolite, and quartz-mica schist from bottom to top. The entire crustal domain subsequently underwent folding, thrusting, and metamorphism during the Pan-African orogeny, peaking at 530 Ma [18].

Emerald mineralization of the Kagem Mine is hosted by the talc-magnetite schist, which contains talc-chlorite-actinolite-magnetite schists (Figure 2) [7,14]. These schists were identified as metamorphosed komatiites—Mg-rich ultramafic rocks [9]. During the late stages of the Pan-African orogeny, the talc-magnetite schist was intruded by beryllium-rich pegmatite dykes (typically 2–10 m thick) [19,20]. These pegmatites appear as feldspar-quartz-muscovite bodies or as minor quartz-tourmaline veins and are believed to be related to neighbouring granitic rocks [7]. The steeply inclined pegmatite dykes and quartz tourmaline veins typically trend north to south or northwest to southeast.

The emerald mineralization in the Kagem Mine results from the interaction between metabasites and pegmatites and their accompanying hydrothermal fluids [9,19]. Emeralds are found in the phlogopite reaction zone (typically 0.5–3 m wide) between the talc-

magnetite schist and the pegmatites and in the quartz-tourmaline veins, which produces gem-quantity emeralds, but only a tiny percentage [3]. In the reaction zone, talc-magnetite schist provided the important chromophore of chromium, while the pegmatites provided the beryllium. Emerald was formed under the proper pressure, temperature conditions (400–600 MPa and 590–630 °C) and chemical environment around 500 Ma [9,21]. The area subsequently underwent intense shearing and folding during the Lufilian orogeny, which may account for the fracturing and opacity of many emerald crystals [7].

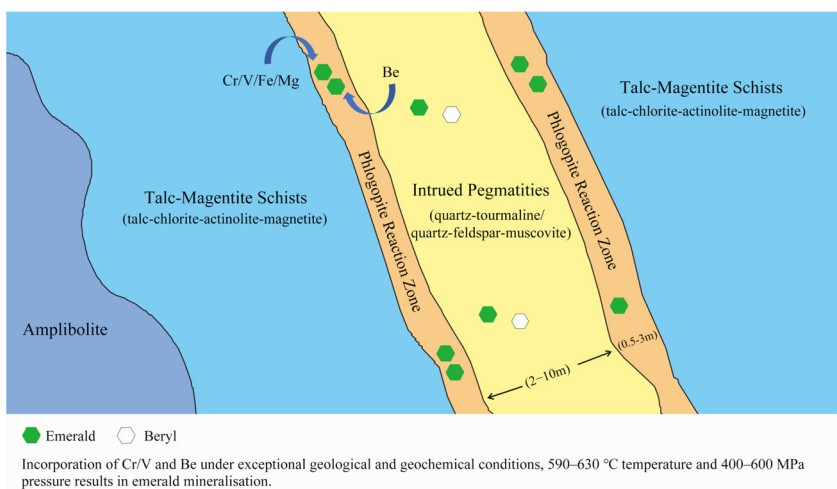

**Figure 2.** Schematic representation of the metasomatic reaction zone between the talc-magnetite schists and the pegmatite formed common beryl and gem-quality emeralds. Modified from [22].

## 3. History

There are two major emerald areas in Zambia, including Musakashi and Kafubu areas. The Musakashi area was first reported in 2005. The emeralds from this area were characterized by their intense bluish-green color and multiphase inclusions, which were similar to the emeralds found in Muzo, Colombia [3,10,23]. Beryl was first discovered in the Kafubu area in 1928 by two geologists named Dicks and Baker. Some small exploration works were carried out during the 1940s and 50s' [7]. Zambia's Geological Survey Department mapped the Miku area and verified the deposit in 1971 [13,14]. After new occurrences were discovered in the 1970s, the Kafubu area became a significant producer of fine-quality emeralds. Because of the dramatic expansion of production and extensive illegal mining, the Government relocated the local population and established a restricted zone called the Ndola Rural Emerald Restricted Area [9]. In 1980, Kagem Mining Ltd. (55% owned by the Zambian Government) was authorized to explore and mine the Kafubu area in the same year [9].

In 2004, the British public-listed Gemfields Resources PLC began systematic exploration near the Pirala mine, south of the Ndola River, and discovered significant emerald deposits. Gemfields was awarded a management contract there in 2007. In the following year, Gemfields acquired 75% ownership of the mine, the remainder being held by the Government [11].

Kagem is primarily an open-pit mine, which presents the advantage of providing accessibility to every carat of emerald (Figure 3). After the surrounding rock is removed, miners utilize hammers and chisels to recover the emeralds. Any extracted production goes into a red production box (Figure 4). Since 2010, the Kagem Mine has been responsible for approximately 50% of emerald production in Zambia. Despite the high production volume, only a small portion of gem-quality emeralds are available for exportation In 2022, the Kagem Mine produced a total of 37.2 million carats of emeralds and beryl, including 259,500 carats of premium emeralds [24].

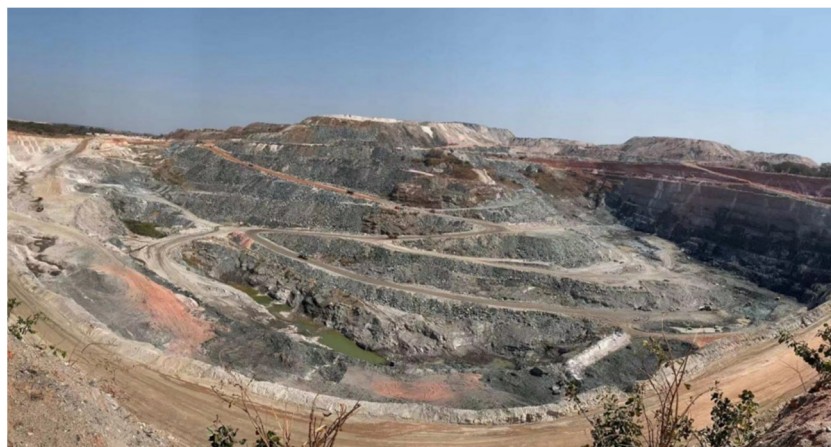

**Figure 3.** The Kagem Mine is the world's largest open-pit mine for emeralds and is located in the Ndola Rural District, Copperbelt Province, Zambia. Photo by Xiangjie Xiao.

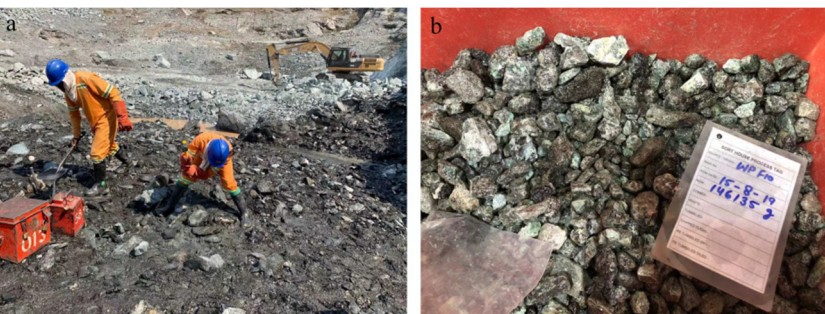

**Figure 4.** (**a**) Miners use hammers and chisels to extract emeralds from the rock in the reaction zone; (**b**) The collected emeralds were in a secure box. Photo by Xiangjie Xiao.

Colored gemstone auctions have become a major source of revenue for the Gemfields Group. The company established its proprietary grading system to assess each gem according to its individual characteristics (size, color, shape and clarity) and implemented a pioneering auction and trading platform. The auctions were divided into two quality ranges: one for higher quality (HQ) emeralds and one for commercial quality (CQ, formerly known as lower-quality before 2016) emeralds [2]. The auctions are held in multiple cities, including Jaipur, Johannesburg, Lusaka, London, Dubai and Singapore [24]. Gemfields had held 43 auctions of emerald and beryl mined at Kagem up to November 2022 and generated $899 million in aggregate revenues [24]. The per-carat price for HQ and CQ emeralds in 43 auctions has been recorded (Figure 5a,b). The specific auction mix and the quality of the lots offered at each auction vary in characteristics such as size, color and clarity due to changes in mined production and market demand. Overall, the price of emeralds showed a clear upward trend, with HQ emeralds increasing from an initial $4.40 per-carat (July 2009) to a peak of $155.90 (May 2022) per-carat and CQ emeralds rising from an initial $0.31 per-carat (January 2010) to a highest $9.37 per-carat (April 2022). The value and demand for emeralds are rising steadily.

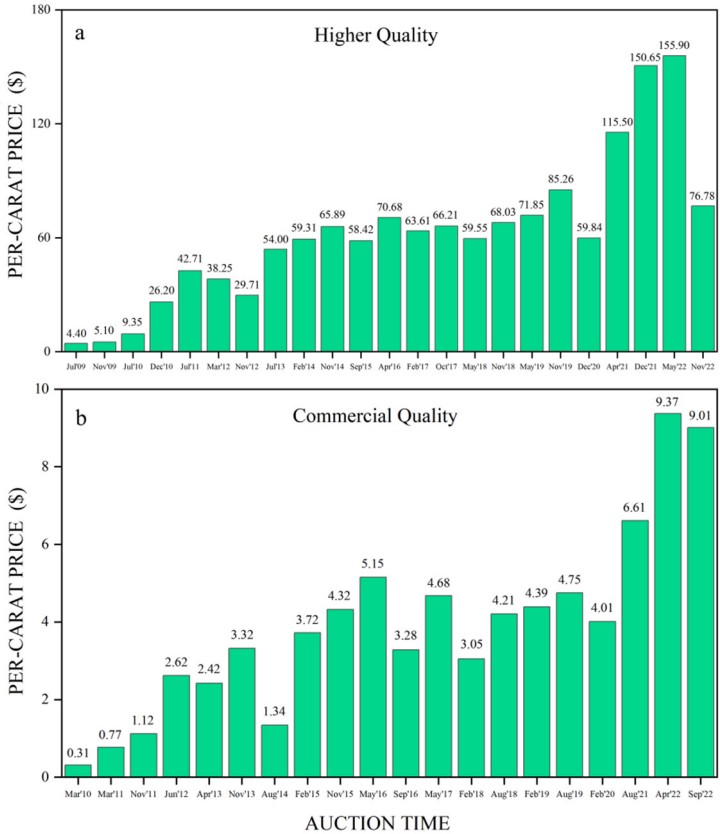

**Figure 5.** (**a**) The per-carat price (calculated as: received/total weight sold) for higher quality emeralds from the Kagem Mine at auctions has been tracked since July 2009; (**b**) The per-carat price for commercial quality emeralds from the Kagem Mine at auctions has been recorded since March 2010. Auction data from [24].

## 4. Materials and Methods

A total of 30 transparent faceted emeralds (K1-30) were acquired from a gem trader who had visited the Kagem Mine and collected them (Figure 6). According to the GIA's classification scheme, the emeralds we obtained belong to E-type samples [25]. There was no visible color zoning in the speciments although few contained some dark inclusions, and a few dark inclusions were visible, and five crystals were treated with oil and wax fillings.

Standard gemological properties were obtained on all the samples. Refractive indices and birefringence were obtained with a gem refractometer (FGR-003A, FABLE, Shenzhen, China). UV fluorescence was examined under a UV lamp with long-wave (365 nm) and short-wave (254 nm) light in a darkened box. We also tested their reaction under the Chelsea filter (FCF-25, FABLE, Shenzhen, China). Dichroism was observed and photographed under a polarizing film (FID-1, FABLE, Shenzhen, China). Specific gravity was determined by the hydrostatic method. Internal features were observed and photographed by a Leica M205A microscopic system. In some cases, a polarizing microscope was used as well.

Inclusions were identified using a JASCO NRS-7500 Series Confocal Raman Microscope (JASCO, Tokyo, Japan) with 532 nm and 457 nm lasers at the Gemmological Institute, China University of Geosciences, Wuhan. Solid inclusions were identified in the 2000–100 $cm^{-1}$ range with the 532 nm laser using a grating of 600 grooves/mm. Two-phase inclusions were identified in the 4000–100 $cm^{-1}$ range with the 457 nm laser using a grating of 600 grooves/mm. The laser power was around 10 mW. Three scans with 15 s integration time for each scan were taken for a single spectrum. The Raman shift was calibrated with monocrystalline silicon (at 521 $cm^{-1}$).

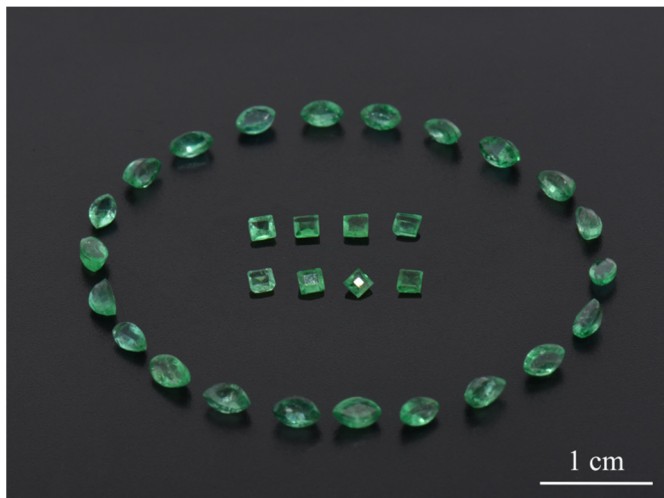

**Figure 6.** Thirty faceted emeralds from the Kagem Mine, Zambia (the average weight was 0.15 ct). Photo by Ran Gao.

UV-Vis-NIR spectra were recorded by a Jasco MSV–5200 micro-spectrometer (JASCO, Tokyo, Japan) in the range of 300–900 nm, at a scan speed of 200 nm/min at the Gemmological Institute, China University of Geosciences, Wuhan. The optical aperture was set at 100 nm and 0.5 nm data interval. The black cross center of the faceted sample, when viewed through a polarizing mirror, indicates the C-axis. Polarized spectra of each oriented sample were collected to obtain ordinary ray (o-ray) and extraordinary ray (e-ray) absorption spectra.

Fourier-transform infrared spectroscopy (FTIR) was performed using a Bruker VER-TEX 80 FTIR spectrometer (BRUKER OPTICS, Billerica, MA, USA), using 32 scans and 4 cm$^{-1}$ spectral resolution at the Gemmological Institute, China University of Geosciences, Wuhan. The scanning ranges were 9000–4000 cm$^{-1}$ in transmission mode.

Trace element contents were analyzed by laser ablation-inductively coupled plasma-mass spectrometry (LA-LCP-MS) using an Agilent 7700 ICP-MS combined with a GeoLas 193 nm laser (Agilent, Singapore) at Wuhan SampleSolution Technology Co., Ltd., Wuhan, China. We set the laser fluence at 9 J/cm$^2$ with a 6 Hz repetition rate and the laser spot size at 44 µm diameter. Each analysis incorporated a background acquisition time of approximately 20–30 s, followed by 50 s of ablation. Element concentrations were calibrated against multiple reference materials (BCR-2G, BHVO-2G, and BIR-1G) without using an internal standard, and Al was chosen as the normalizing element [26]. Standard reference material NIST 610 glass was also applied to time-drift correction. The standard for LA-ICP-MS measurements is that the calibration values of the monitored reference materials agree within the error range within the recommended values. Quality control deviations: Major elements within 5% uncertainty and trace elements within 10% uncertainty. Two to three spots per sample were analyzed.

## 5. Results

### 5.1. Gemological Properties

The gemological properties of emeralds from the Kagem Mine are summarized in Table 1. The emeralds ranged from green to blueish green, and some displayed an attractive saturated bluish-green color. The refractive indices varied from 1.578–1.591 for $n_e$ and 1.589–1.597 for $n_o$ with birefringence between 0.004 and 0.008. These values are higher than most of the significant emerald deposits. Specific gravity values ranged from 2.67 to 2.86. The emeralds from the Kagem Mine were typically inert to long- and short-wave UV radiation. The emeralds showed no reaction under the Chelsea filter. Dichroism was medium to strong yellowish green (o-ray) and bluish green (e-ray).

**Table 1.** Gemological properties of emeralds from Kagem Mine, Zambia.

| | |
|---|---|
| Color | Light to medium bluish green; typically, a saturated green with a medium tone |
| Clarity | Very slightly to heavily included |
| Refractive indices | $n_o$ = 1.589–1.597; $n_e$ = 1.578–1.591 |
| Birefringence | 0.004–0.008 |
| Specific gravity | 2.67–2.86 |
| Pleochroism | Medium to strong yellowish green (o-ray) and bluish green (e-ray) |
| Fluorescence | Inert to long- and short-wave UV radiation |
| Chelsea filter | No reaction |
| Visible Spectrum | Distinct lines at~680 nm; and complete absorption <430 nm |
| Internal features | Two-phase inclusions with a gas bubble, display elongated, hexagonal, oval, or irregular shapes, were common |
| | Densely distributed black mineral crystals, usually with a hexagonal or rectangular outline |
| | Colorless transparent crystals |
| | Strong iridescent colors in fissures |
| | Mineral inclusions: needle-like actinolite; clusters of black magnesite; colorless dolomite; jagged or oval graphite encased in actinolite |

### 5.2. Microscopic Characteristics

Two-phase inclusions with a gas bubble were commonly found in the Kagem emeralds, displaying various shapes: rectangular, elongated, hexagonal, oval, or irregular shapes (Figure 7a). Some two-phase inclusions occurred as isolated negative crystals along a healed fissure plane, with an incomplete hexagonal prism shape (Figure 7b). At room temperature, the gas bubbles typically occupy approximately one-third of the volume of the cavity hosting the two-phase inclusions. Raman analysis confirmed that the liquid phase of the two-phase inclusions primarily consisted of water or a liquid mixture of $H_2O$ and $CO_2$. The gas components included $CO_2$ + $N_2$ or $CO_2$ + $N_2$ + $CH_4$ (Figure 8). There was a two-phase inclusion with liquid and solid phases, and the latter was identified as dolomite (peaks at 174, 296, and 1095 cm$^{-1}$) by Raman analysis (arrow 1 in Figure 7a). It is worth noting that a hexagonal multi-phase inclusion, containing a gas bubble and an obvious solid phase in an aqueous solution, is rare in Kagem emeralds (arrow 2 in Figure 7a).

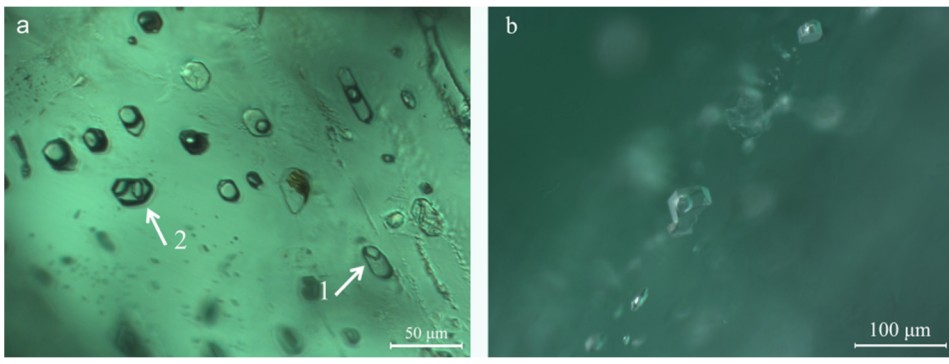

**Figure 7.** (**a**) Two-phase inclusions with various shapes, a two-phase inclusion with solid phase and a hexagonal three-phase inclusion (indicated by white arrows 1, 2); (**b**) Two-phase inclusions appear as negative crystals with an incomplete hexagonal prism. Photomicrographs by Ran Gao.

Various mineral inclusions were observed in Kagem emeralds under the standard gemological microscope. Colorless needle-like minerals of varying lengths were seen in many samples and identified as actinolite through Raman analysis (Figure 9a). Black jagged minerals, identified by Raman analysis as graphite, were visible in the actinolite needles (Figure 9a). Occasionally, graphite appeared in oval or irregular shapes. Densely distributed black mineral crystals (Figure 9b), which appeared hexagonal or rectangular under high magnification, were identified using Raman spectroscopy as magnetite. Similar

magnetite inclusions were also found in emeralds from Davdar, China [27]. A colorless transparent crystal was identified as dolomite via Raman spectroscopy (Figure 9c). Partly healed fissures displayed intense iridescent colors (Figure 9d).

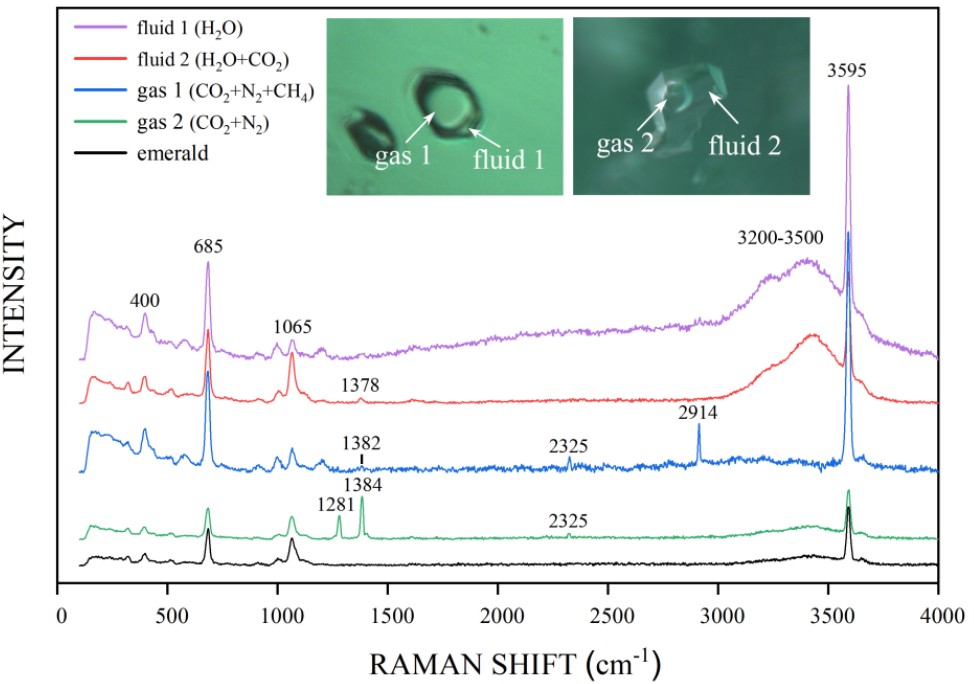

**Figure 8.** Two-phase inclusions with a gas bubble in emeralds from Kagem Mine. Emerald: peaks at 400, 685 and 1065 cm$^{-1}$; $CO_2$: peaks at 1281 and 1378–1384 cm$^{-1}$; $N_2$: peak at 2325 cm$^{-1}$; $CH_4$: peak at 2914 cm$^{-1}$; $H_2O$ in the inclusion: bands from 3200–3500 cm$^{-1}$; $H_2O$ in structure channel: peak at 3595 cm$^{-1}$ [28,29]. Spectra are offset vertically for clarity. Photomicrographs by Ran Gao.

*5.3. UV-Vis-NIR Spectroscopy*

Representative UV-Vis-NIR absorption spectra of the emeralds from the Kagem Mine are illustrated in Figure 10. Although the intensities varied among the samples, all exhibited the same bands for the ordinary and extraordinary rays. Specifically, the ordinary ray (o-ray) showed bands at 372, 430, 610, 637, and 840 nm, as well as a doublet at 681 and 684 nm. The bands at 430, 610, and 840 nm were broad and the positions were estimated. The extraordinary ray (e-ray) displayed bands at 372, 427, 625, 640, 662, 684, and 840 nm, the absorbance of a peak at 372 nm was weaker than its o-ray counterpart.

The bands at 430 and 610 nm (o-ray) and 427, 625 and 640 nm (e-ray), as well as the peak at 684 nm (e-ray), indicated the presence of $Cr^{3+}$, which causes the green color in emerald [30,31]. Additional, weaker peaks at 637, 662 and the doublet at 681 and 684 nm were also associated with the presence of $Cr^{3+}$. The broad absorption bands from 600 to 750 nm were possibly linked to $Fe^{3+}$-$Fe^{2+}$ charge transfer [32]. Moreover, the peak at 372 nm indicated the presence of $Fe^{3+}$, and the band at 840 nm was caused by $Fe^{2+}$ [3,10,30,33], showing weaker intensity than the bands caused by $Cr^{3+}$. The UV-Vis-NIR spectra feature differs from some previous [10,34,35], but is similar to other scholars' findings [3,36].

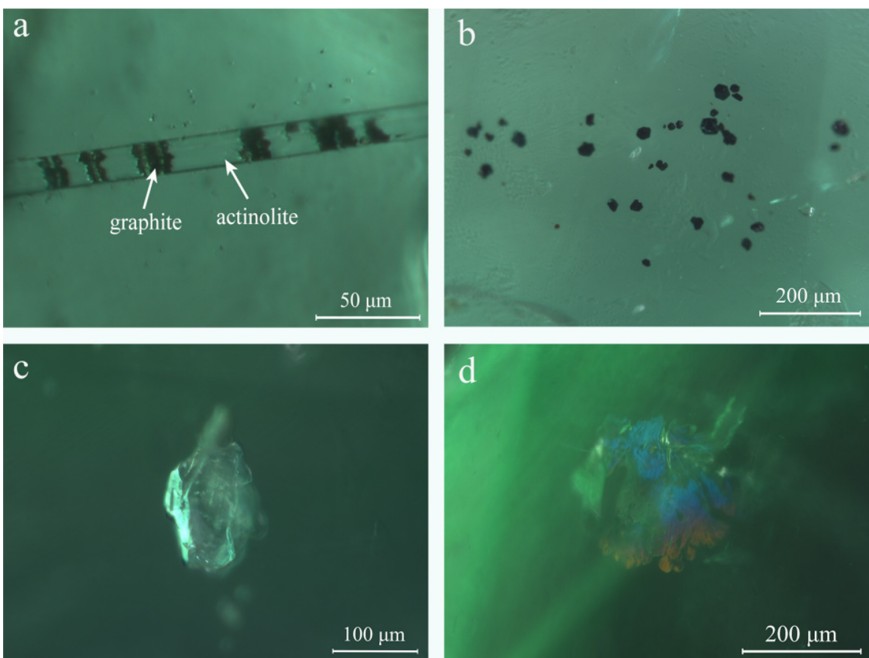

**Figure 9.** Various mineral inclusions were observed in Kagem emeralds. (**a**) Black jagged graphites were wrapped inside a colorless needle-like growth tube, identified using Raman spectroscopy as actinolite; (**b**) Cluster of black magnetite appeared in the form of hexagons or rectangles under high magnification; (**c**) An isolated colorless crystal was consistent with the Raman spectrum of dolomite; (**d**) Obvious iridescent colors in partly healed fissures. Photomicrographs by Ran Gao.

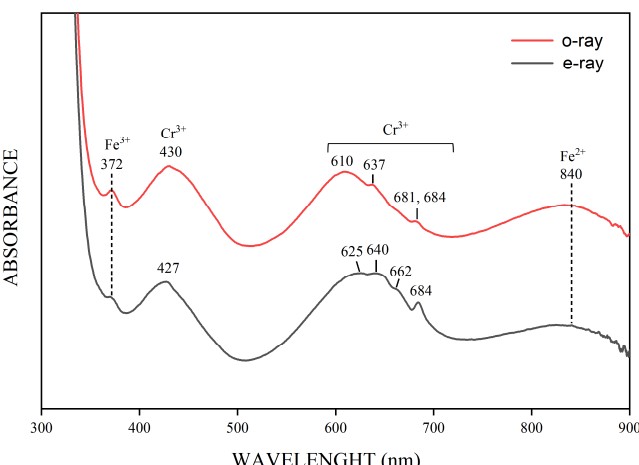

**Figure 10.** Representative UV-Vis-NIR spectra were collected on sample K-3, which has 2652 ppm Cr, 267 ppm V, and 8759 ppm Fe. The spectra indicate the presence of $Cr^{3+}$, $Fe^{2+}$, and $Fe^{3+}$. The intensities or positions of the Chromophore elements varied in different spectral orientations. The spectra are offset vertically for clarity.

### 5.4. Infrared Spectrometry

The near-infrared (NIR) spectrum of the emeralds was mainly related to the existing mode of $H_2O$ molecules in the channel (Figure 11). $H_2O$ molecules in the c-channels have been mainly classified as type I or type II water with their twofold axis perpendicular or parallel to the crystal c-axis, respectively [37].

In the NIR range of 8000–4000 $cm^{-1}$, the absorption spectrum bands of emeralds are predominantly attributed to the combined frequency and double-frequency vibration of structural water [38]. The most obvious band at 5274 $cm^{-1}$ was caused by the combined

frequency of bending ($v_2$) and antisymmetric stretching ($v_3$) modes of type I or II $H_2O$ molecules [38,39]. The sharp bands at 7097–7075 cm$^{-1}$ doublet, slightly weak bands at 6840 and 7268 cm$^{-1}$, and two small shoulder bands at 5340 and 5205 cm$^{-1}$ near the 5274 cm$^{-1}$ band were assigned to type II $H_2O$ molecules, which were associated with alkali ions in the channels of the emerald structure. The relatively strong band at 7140 cm$^{-1}$ was related to the overtone frequency absorption of type I water, whereas other spectral bands of type I water, such as 7275 and 6820 cm$^{-1}$ [38,40], were too weak to be discernible.

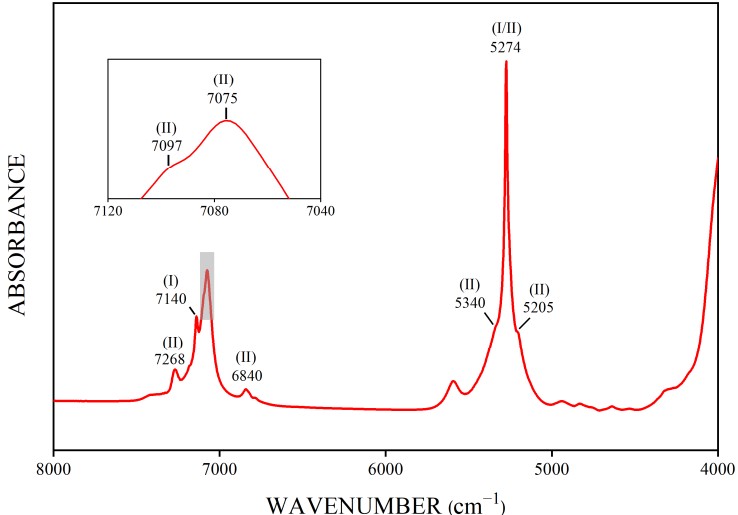

**Figure 11.** The representative FTIR spectrum of Kagem emeralds shows some significant bands caused by the vibrations of type I or type II $H_2O$ molecules in the channels of the emerald structure. This graph displays that the spectral band intensities of type II $H_2O$ molecules are higher than those of type I $H_2O$ molecules in Kagem emeralds.

*5.5. Trace Element Analysis*

Twenty faceted emeralds from the Kagem Mine were analyzed by LA-ICP-MS, three spots for each sample and calculated their average values. The results are shown in Table 2.

**Table 2.** Chemical composition (in ppm) of Kagem emeralds, obtained by LA-ICP-MS.

| Samples | Element | Min–Max | Average (SD) | Median | LOD |
|---|---|---|---|---|---|
| | Li | 105.2–250.1 | 164.4 (31.46) | 165.6 | 1.05 |
| | Na | 13,810–17,410 | 16,440 (730.0) | 16,640 | 23.89 |
| | Mg | 13,350–17,230 | 15,650 (725.3) | 15,730 | 4.17 |
| | K | 121.0–283.9 | 184.9 (35.83) | 179.2 | 25.99 |
| | Sc | 96.78–1534 | 435.8 (298.6) | 340.0 | 0.49 |
| | Ti | 3.967–51.28 | 16.17 (8.190) | 14.75 | 1.07 |
| | V | 196.8–528.8 | 307.2 (77.95) | 293.1 | 0.25 |
| Kagem emeralds | Cr | 539.6–4844 | 2034 (1010) | 1758 | 6.26 |
| 30 samples | Fe | 4822–14,160 | 9713 (2181) | 9875 | 59.31 |
| | Ni | 6.845–37.43 | 19.49 (6.771) | 20.22 | 5.61 |
| | Zn | BDL–15.86 | 3.766 (3.608) | 3.225 | 2.02 |
| | Ga | 2.916–50.49 | 31.15 (10.02) | 34.10 | 0.40 |
| | Rb | 9.381–20.78 | 14.29 (2.246) | 14.29 | 0.75 |
| | Cs | 128.4–1052 | 567.3 (268.6) | 503.6 | 0.37 |

Data was rounded to 4 Significant Figures; SD = standard deviations; LOD = limit of detection; BDL = below detection limit.

The Kagem emeralds for this study tended to have abundant alkali metals Li, Na, K, Rb, and Cs. Total alkali element concentrations ranged from 14,227 to 18,634 ppm (avg. 17,405 ppm). The dominant alkali metal was Na, which ranged from 13,812 to 17,408 ppm,

averaging 16,437 ppm. The emeralds also contained a significant amount of Mg and Fe, ranging from 13,348 to 17,225 ppm (avg. 15,645 ppm) and from 4822 to 14,159 ppm (avg. 9713 ppm), respectively. Compared to the concentrations of Sc we measured (97–1534 ppm, avg. 436 ppm), analyses of emeralds by Saeseaw et al. (2014) [10] display relatively low Sc (12–75 ppm, avg. 31 ppm) in emeralds from Kafubu area. Cr (540–4844 ppm, avg. 2034 ppm), the most crucial chromophore in emeralds, was higher than V (197–529 ppm, avg. 307 ppm).

## 6. Discussion

### 6.1. Inclusion Characteristics of Kagem Emeralds

The rectangular and hexagonal shapes of the two-phase inclusions found in Kagem emerald were consistent with the multi-phase inclusion characteristics observed in schist-hosted emerald deposits, such as those found in Brazil, Russia, and most African deposits. We updated the component of the gas phase in two-phase inclusions. It is not monolithic and can contain not only $CO_2$ but also occasionally $N_2$ and $CH_4$. Multi-phase inclusions with a solid phase were rare. $H_2O$-NaCl-$CO_2$ type inclusions were not found in our samples. Black graphite encased in actinolite was first observed in Kagem emeralds. The occurrence of abundant black magnetite and transparent dolomite further demonstrates a direct association between the internal characteristics of gemstones and their geological background. The talc-magnetite schist contained high concentrations of Fe and Mg.

### 6.2. $Fe^{2+}$ Absorption Band in UV-Nis-NIR Spectroscopy

The Fe content in Table 2 represents the total Fe acquired by LA-ICP-MS, including $Fe^{3+}$ and $Fe^{2+}$. Kagem emeralds showed the weaker $Fe^{2+}$-related absorption band in the NIR region compared to the bands caused by $Cr^{3+}$. However, a lack of correlation was found between the intensity of the $Fe^{2+}$-related band and the total Fe content [41]. Divalent metal ions ($Me^{2+}$), such as $Mg^{2+}$ and $Fe^{2+}$, can substitute for $Al^{3+}$ in the emerald structure. In order to maintain charge balance, the channels must be occupied by alkali ions such as $Na^+$, $K^+$, $Rb^+$, and $Cs^+$ (mainly $Na^+$) [1,42], as follows:

$$Al^{3+} = Me^{2+} + Na^+, \tag{1}$$

Moreover, the linear positive correlation between $Na^+$ content and the sum of divalent cations is approximately 1:1 [41], thereby $Fe^{2+}$ concentration can be calculated by the following equations:

$$Na^+{}_{ppma} = Fe^{2+}{}_{ppma} + Mg^{2+}{}_{ppma}, \tag{2}$$

$$X_{ppma} = X_{ppm} \frac{E_{am}}{X_{am}} \tag{3}$$

To clarify, ppma represents parts per million by atom, $E_{am}$ represents the average atomic mass of emerald chemical formula, and $X_{am}$ represents the atomic mass of elements (Na, Mg and Fe).

The proportion of $Fe^{2+}$ in the total Fe we calculated varied from 5.37% to 59.33%. A part of $Fe^{2+}$ generated intervalence charge transfer with $Fe^{3+}$, and this portion of $Fe^{2+}$ did not affect the band in the NIR region. Additionally, the content of $Cr^{3+}$ was a crucial factor to consider. We found a positive relationship between the $Fe^{2+}$ to $Cr^{3+}$ content ratio and the intensity ratio of the $Fe^{2+}$-related band at 840 nm to the $Cr^{3+}$-related band at 610 nm (o-ray). We speculated that the weaker band of $Fe^{2+}$ in our samples was due to the low $Fe^{2+}$ to $Cr^{3+}$ value ratio. The concentration of $Fe^{2+}$ can be roughly determined based on the content of bivalent cations and alkali metals. However, when discussing the relative strength of the $Fe^{2+}$-related absorption band, the concentration of $Cr^{3+}$ and $V^{3+}$ should also be considered.

### 6.3. Channel Water Types of Kagem Emerald

Alkali metal ions, water and other macromolecules that exist within the structural channel of emeralds can maintain the charge balance and structural stability of minerals. The concentration of alkali metal ions affects the type of emerald channel water. In the presence of alkali ions, the axis of symmetry of adjacent $H_2O$ molecules is changed from perpendicular to parallel to the c-axis due to the electrostatic attraction between the charged cation and the oxygen of the $H_2O$ molecule [39].

Kagem emeralds, dominated by Na, contain abundant alkali metal ions (avg. 17,405 ppm). The NIR spectra of Kagem emeralds (Figure 11) revealed that the bands for type II $H_2O$ at 7097–7075 cm$^{-1}$ doublet were more intense than those for type I $H_2O$ at 7140 cm$^{-1}$. Moreover, the shoulder bands at 5340 and 5205 cm$^{-1}$ were closer to 5274 cm$^{-1}$ and were more obvious with higher proportions of type II $H_2O$. As mentioned above, Kagem emeralds dominate type II $H_2O$ and the Kagem Mine belongs to one of three mineralization environment types—alkali-rich type (other types include alkali-poor and transition) [38]. Similar emeralds were found in Swat Valley, Pakistan, Goias, Brazil, and most African deposits.

### 6.4. Origin Traceability Analysis

A log-log plot of Rb versus Cs content can separate the Kagem mine from emeralds of all other significant deposits (Figure 12). Kagem emeralds exhibit low-to-moderate Rb values and moderate-to-high Cs, with a slight overlap with emeralds from Ural, Russia. Emeralds from Malipo, China, and Chitral, Pakistan display high Cs content, while Zimbabwean and Madagascan samples exhibit high Rb value. The emeralds from Colombia show very low Rb and Cs values. The plotted points of the two China deposits can be easily separated by their Cs content.

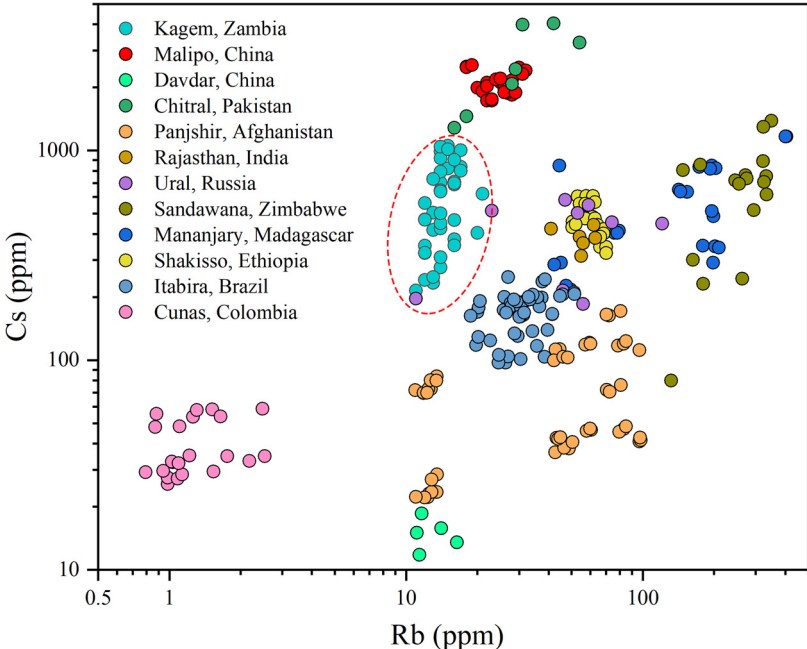

**Figure 12.** A log-log plot displays the Rb versus Cs contents in emeralds from twelve different localities. Kagem emeralds (within the red dashed) show moderate-to-high Cs values and low-to-moderate Rb. Other sources are from [42–51].

A better separation between the Kagem Mine and other significant emerald localities can be made by plotting Na versus K (Figure 13). Kagem emeralds display high Na values combined with low-to-moderate K, despite a slight overlap with emeralds from Sandawana, Zimbabwe and Shakisso, Ethiopia, which have relatively high K. Emeralds from Madagascar, Ethiopia, Zimbabwe, and India, all of which belong to Type IA occurrences [1], contain

high Na value over 10,000 ppm. The emeralds from Panjshir, Afghanistan exhibit a broad range of Na and K values in this plot. A log-log plot of Fe versus Zn was created to better distinguish our samples from those in Zimbabwe and Ethiopia (Figure 14). The samples from Kagem show relatively low Zn value.

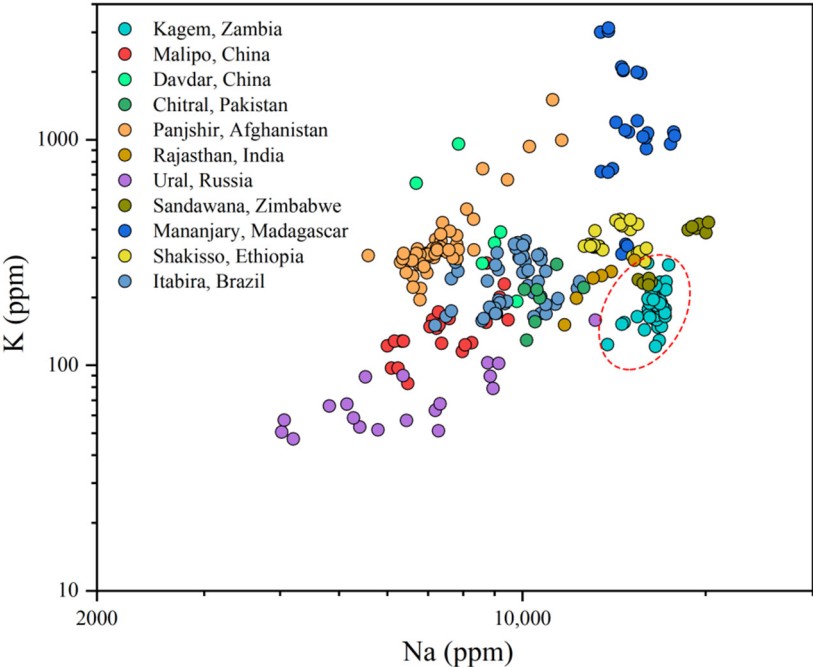

**Figure 13.** A Na vs K plot shows the relatively high Na and low-to-moderate K concentrations in Kagem emeralds (within the red dashed) compared to the other significant deposits. Sources for some of the data are the same as in Figure 12.

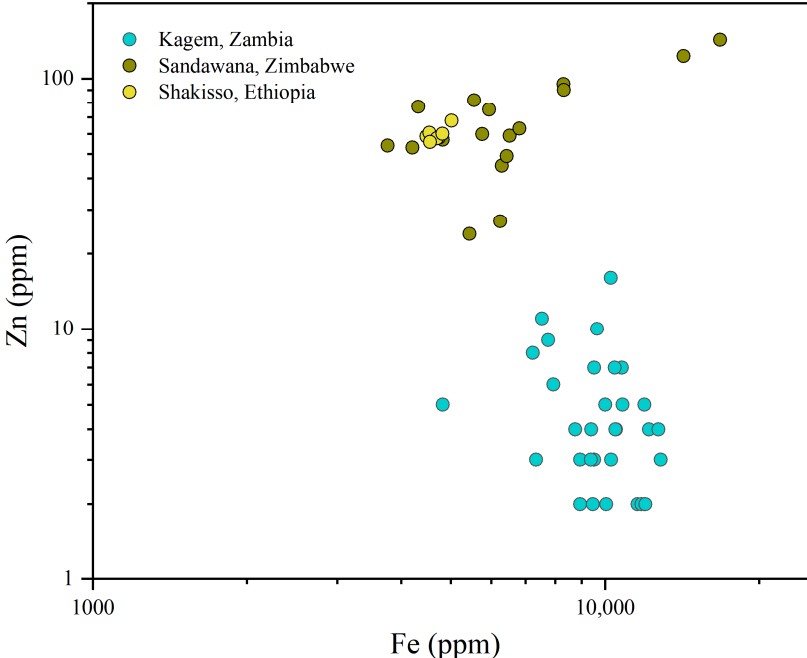

**Figure 14.** The log binary graph of Fe and Zn shows that Kagem emeralds are not similar to those from Zimbabwe and Ethiopia in this data and can be distinguished. Data source: [43,51].



## 7. Conclusions

The origin tracing of emeralds requires sufficient samples as support, and new features have been discovered with the advancement of research methods. On the basis of previous research, this study provides new discoveries on Kagem emeralds, including the inclusion, spectral characteristics and chemical analyses.

Two-phase inclusions are common in Kagem emeralds, with the fluid phase containing $H_2O$ or liquid mixtures of $H_2O + CO_2$, and the gas phase consisting of combinations of $CO_2 + N_2$ or $CO_2 + N_2 + CH_4$. A special two-phase inclusion with a solid phase has been identified as dolomite. Raman spectroscopy analysis revealed that Kagem emeralds contain a variety of solid inclusions, including actinolite, graphite, magnetite, and dolomite. Kagem emeralds typically show a conspicuous band caused by $Fe^{2+}$ in the NIR region, and its relative intensity is affected by various elements. Kagem emeralds contain moderate-to-high Cs content (avg. 567 ppm) and low-to-moderate Rb (avg. 14 ppm) content. Additionally, they tend to exhibit high Na values (avg. 16,440 ppm) combined with low-to-moderate K (avg. 185 ppm). The projection diagram of Rb vs. Cs, Na vs. K, and Fe vs. Zn can distinguish the Kagem Mine from other significant emeralds deposits.

**Author Contributions:** Conceptualization, R.G. and Q.C.; formal analysis, R.G.; investigation, R.G. and Q.C.; writing—original draft preparation, R.G.; writing—review and editing, Q.C., Y.L. and H.H.; supervision, Q.C. All authors have read and agreed to the published version of the manuscript.

**Funding:** This research was funded by the Hubei Gem & Jewelry Engineering Technology Center (No. CIGTXM-03-202201 and No. CIGTXM-04-S202107), the Fundamental Research Funds for National University, China University of Geosciences (Wuhan) (No. CUGDCJJ202221), the Philosophy and Social Science Foundation of Hubei Province (No. 21G007), NSFC funding (No. 41874105).

**Data Availability Statement:** All supporting data and computational details are available on written request. These data are stored by the main author of this article.

**Acknowledgments:** We are grateful to Xiangjie Xiao for providing emerald samples and photographs of the deposit.

**Conflicts of Interest:** The authors declare no conflict of interest.

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
