# Peer review of "Update on Emeralds from Kagem Mine, Kafubu Area, Zambia"

_minerals, doi:10.3390/min13101260_

Round 1
Reviewer 1 Report (Previous Reviewer 1)
See the annotated manuscript in the attachment

Author Response
Please see the attachment.

Reviewer 2 Report (Previous Reviewer 2)
Thank you very much for giving me a second opportunity to review the revised manuscript. To my sense the manuscript is much improved and can be accepted for publication.
Author Response
Thank you for your support and attention. Your guidance has been invaluable to my research, and I will revise this manuscript carefully.
Reviewer 3 Report (New Reviewer)
There are two major critics that I have to express:
1. The samples in Figure 1 look very much alike, meaning they are uniform in colour. I have seen much more diversity in Zambian samples. I assume that the uniformity of spectra and chemistry is due to this uniformity, and therefore the results do not represent a characterisation of emeralds from the Kafubu mine. It would have been better to have a representative variety of colours.
2. All comparison data are taken from the literature. It would have been good to see data from the authors that have been taken with the same instruments to make sure the parameters are the same.
Another point of critics is the discussion chapter about UV-Vis spectra. It appears either not too well researched, or not clearly outlined. Some of the comments are contradictive.

The grammar is not always right, and the authors use the present and the past randomly. This can be improved. A few examples are given in the comments of the text.
Author Response
Please see the attachment. Please ignore the watermark in the PDF.

Reviewer 4 Report (New Reviewer)
This paper written on the Emerals from Zambia is well written and well presented although few changes shuld be carried out before pubblication. The title is nice and short and clearly indicates the content of the paper. Overall, the paper is well constructed in terms of data but there are a couple of section that should be moved and some info seems too details and irrelevant for the paper. Too many figures are submitted for the size of the paper and indeed some can be deleted as totally not necessary as they are not even mentioned in the text. The abstract is well written but the aim/achievement of the work I beleive is missing. The discussion are well written and conclusions are short but ok. The references list seems appropriate.
More comments and/or suggestions for improvement are added to the main body text file.

Good English! with very little errors.
Round 2
Reviewer 3 Report (New Reviewer)
in Kagem mine is a result ofMine results from the inter- 146: in the Kagem mine
s solution, was rare in Kagem 232: Is it not rare anymore?
appeared as hexagonal or rectangular under high magni- 240: If you use the adjective you have to delete the as.
6.3. Channel water types of Kagem emerald 355: Here it would be nice to see an actual spectrum.
Author Response
in Kagem mine is a result of Mine results from the inter- 146: in the Kagem mine
Thanks for your correction.
s solution, was rare in Kagem 232: Is it not rare anymore?
Sorry for that. I used the wrong tenses of the word.
appeared as hexagonal or rectangular under high magni- 240: If you use the adjective you have to delete the as.
Thanks for your suggestion. We deleted the as.
6.3. Channel water types of Kagem emerald 355: Here it would be nice to see an actual spectrum.
The content of 6.3. (Channel water types of Kagem emerald) is a discussion of section 5.4 (Infrared Spectrometry), and its corresponding spectrum is shown in Figure 12. We have marked it in section 6.3.
In addition, we simplify our conclusions.
This manuscript is a resubmission of an earlier submission. The following is a list of the peer review reports and author responses from that submission.
Round 1
Reviewer 1 Report
This paper provided information and results are useful.I have a number of comments about wording and about details as follows:
Line 31 The Kafubu area also includes the latest Kamakanga deposit. See Zhang Yi et.al., 2023 for more details.
Line 94 Please indicate references or sources for auction data in the article.
Line 168 Muscovite and phlogopite are described in the geological context, but biotite appears in Fig. 6. Please confirm exactly which mica?
Line 171 Please add the sample number and keep the number uniquely in this article.
Line 171 Please indicate the source of the samples, who provided the samples received? Can you ensure that they are from the Kagem mine?
Line 188 Why was the wavelength of 532 nm chosen for solid inclusions? Why was the wavelength of 457 nm chosen for two-phase inclusions?
Line 217 This article is only the results, not the discussion. Each test result can be supplemented by some analysis and discussion as appropriate.
Line 227 Delete the “Kafubu are”
Line 237 which kind of carbonate?
Line 258 Is it reasonable that gas 1 lacks the absorption peak of 1281 but is considered to be CO2?
Line 284 The absorption of Fe2+ in Figure 11 is lower than that of Cr3+, whereas the UV-Vis absorption spectrum of kafubu emeralds would normally have a significantly higher intensity of Fe2+ than that of Cr3+. Please confirm that this absorption spectrum is representative of the typical absorption spectrum of emeralds from the Kagem mine?
Line 308 Why are there only ten samples of data in Table 2?
Line 321 Add the detection limit.
Line 338 Are there only data for this study in Figure 13? Are there data on the chemical composition of the Kafubu or Kagem mine from previous studies? Do the data from this paper's study overlap with those from previous studies? Is it distinguishable from minor Zambian deposits (e.g. kamakanga, grizzly and chantete)?
Line 342 Figures 13 and 14 do not distinguish well among origins, and the data for Zambia even partially overlap with those for Zimbabwe, Pakistan and Brazil.
Line 347 This is not a conclusion of your research and it is recommended to delete.
English can be improved.
Reviewer 2 Report
Dear editor;
Great thanks for giving me this chance to review the manuscript entilited (Update on Emeralds from Kagem Mine, Kafubu Area, Zambia). I have reviewed this manuscript that contain significant data and provides a full set of data through standard gemological properties, inclusion scenes, color characteristics, and advanced spectroscopic and chemical analyses including Raman, micro micro-UV-Vis-NIR, FTIR, and LA-ICP-MS. I suggest that this manuscript can be published after major revision for the following reasons:
General comments:
· Abstract lacks from the main problem and it is solve. The authors summarized the results of inclusion. It must be rewritten.
· The introduction section must contain one paragraph about emerald mineralization before Kagem mine, Zambia.
· The author said they have received the examined samples from Zambia, How? All the current author from China.
· Please, remove section 3. What is the benefit from Auction section?
· Section 4 (geology setting) is very poor. Please, remove lines from 147 to 168 with figure 6 to the end of paper with new subtitle genesis of emerald.
· Sikait area.
· Please, compare your data with others emerald from different localities such as Sakiat area. https://doi.org/10.1016/j.pce.2022.103266
Minor editing of English language required
Reviewer 3 Report
No new or exciting information on Kagem emeralds, all has been said in G&G 2005. In my opinion, there is enough material published on the Zambian deposits, and the actual minor aspects are not worth to be published as new scientific success in an international paper.